# Evaluation of the Activity of Estragole and 2-Isopropylphenol, Phenolic Compounds Present in *Cistus ladanifer*

Elena Requesón [1], Dolores Osuna [2], Ana del Rosario Santiago [2] and Teresa Sosa [1,*]

1   Department of Plant Biology, Ecology and Earth Sciences, Faculty of Science, University of Extremadura, 06006 Badajoz, Spain; erequeso@alumnos.unex.es
2   Center for Scientific and Technological Research of Extremadura (CICYTEX), Department of Crop Protection, 06187 Badajoz, Spain; mariadolores.osuna@juntaex.es (D.O.); anarosario.santiago@juntaex.es (A.d.R.S.)
*   Correspondence: tesosa@unex.es

**Abstract:** A large number of studies of *Cistus ladanifer* highlight this Mediterranean shrub as a source of the phenolic compounds responsible for the allelopathic potential of this species. There are few phenolic compounds present in *C. ladanifer* that have not yet been studied. The objective of this work was to evaluate the activity of estragole and 2-isopropylphenol on filter paper and soil on monocotyledons (*Allium cepa*) and dicotyledons (*Lactuca sativa*). The results showed that when the test was carried out on paper, the germination and the growth of the *L. sativa* was strongly inhibited by 2 isopropylphenol and estragole. 2 isopropylphenol showed an $IC_{50}$ on the germination of 0.7 mM and 0.1 mM on the germination rate, 0.4 mM on the size of radicle and 0.3 mM on the size of hypocotyl. Estragole showed an IC50 on the germination rate of 1.5 mM and 1.1 mM on the size of hypocotyl. The effects of these pure compounds on *A. cepa* were lower, and when the assays were performed on the soil, they were dissipated. The mixture of these compounds on *A. cepa* had 0.6 mM $IC_{50}$ for the length hypocotyl on paper and 1.1 mM for the length of the radicle on soil. The mixture on *L. sativa* also inhibited the length of the radicle with an $IC_{50}$ of 0.6 mM. On the other hand, it was also observed that estragole stimulated the growth of the *A. cepa* radicle length on soil, showing a hormetic effect with an $EC_{50}$ of 0.1 mM. In conclusion, it can be said that for a species to be allelopathic in nature, it is essential to verify the effect of its possible allelochemicals on the target species, on the soil in which they will exert their action and at the concentrations found in their usual environment, in addition to taking into account the interaction with other compounds present in the medium.

**Keywords:** phytotoxicity; phenolic compounds; allelopathy; hormetic effect; *Cistus ladanifer*; bioherbicides





## 1. Introduction

Plants show a vast variety of secondary metabolites, many of which with allelopathic effects. The way in which these compounds act is similar to that of synthetic herbicides, and their high variety offers the possibility to generate new sources of environmentally friendly herbicides with new mechanisms of action and unexplored target places [1]. In fact, the phenomena of allelopathy and phytotoxic interactions among plants mediated by allelochemicals are already being used in different ways, such as crop rotation, cover crops, dead mulches and aqueous extracts [2]. Phenolic compounds are widely known for being allelopathic [3]. It could be asserted that the phytotoxicity of phenolic compounds is well established, and therefore, they can be an important source of new herbicides [2,4,5].

In the Mediterranean region, the rockrose occupies large areas, often forming almost monospecific communities. It has been demonstrated that the allelopathic effect of *Cistus ladanifer* hinders the establishment of some shrubs and can reduce the area occupied by numerous herbaceous species, which can consequently influence the composition and structure of the communities in which this species is present [6]. The identification and

evaluation of secondary metabolites with phytotoxic activity in leaves of *Cistus ladanifer* supports the relevance of allelopathy for the success of this species in this ecosystem [7–12]. The literature shows that some of these phytotoxic compounds are phenolic compounds and can prevent or delay germination and exert a negative influence on the radicles and hypocotyls of seedlings [6,7,9,10,13]. Such a combined allelopathic effect, i.e., reduction and/or delay of germination, along with the underdevelopment of the seedling, should have fatal consequences for the establishment of plants in semiarid Mediterranean environments [14], which would explain the reduction in the richness and diversity of herbaceous species in Mediterranean communities where it is present [15–17]. The aim of this study was to thoroughly evaluate the phytotoxic activity of phenolic compounds present in *Cistus ladanifer* that have not yet been evaluated and that may present suitable characteristics for use as bioherbicides. Few phenolic compounds remain present in *C. ladanifer* that have not yet been evaluated in isolation [7,8,12]. Two of them are estragole and 2-isopropylphenol, which are present in its leaves. These secondary metabolites are minor compounds from *Cistus ladanifer* essential oil [18–22]. Although these metabolites are also synthesized by other species, it is in *Cistus ladanifer* where we can find them together. In *Cistus ladanifer*, more than 350 compounds from different families have been identified. Research indicates that the allelopathic effect of this species is due to the combination of several compounds [8–10]. Additionally, this study of all the components present in this species can be revealing.

Estragole, 4-allylanisole or 1-methoxy-4-prop-2-enylbenzene, is found in many plant species, particularly in spices, and it is used in perfumes and as a flavoring in foods and liquors. This compound was first isolated from avocado rind (*Persea gratissima* Garth). It is the main compound of tarragon oil (*Artemisia dracunculus* L.), and its presence in a proportion of 60–75% can be the cause of the allelopathic activity of this species [23,24]. If released to the soil, estragole is expected to have low mobility. It exists as a liquid under environmental conditions; therefore, estragole may volatilize from dry soil. Utilizing the OECD 301F ready Biodegradability test, 48% O2 consumption was reached in 4 weeks, indicating that biodegradation may be an important environmental fate process in soil or water [25,26]. Despite all this information, the phytotoxic activity of this compound has never before been evaluated in isolation. For 2-isopropylphenol, *o*-cumenol or 2-propan-2-ylphenol, no information was found relative to these parameters [27], but the phenolic compounds of low molecular weight may have similar biodegradable characteristics.

To carry out an evaluation of these compounds with possible phytotoxic activity, there are standardized bioassays that allow evaluating the degree of affectation that a chemical substance produces in selected test organisms [28,29]. This implies evaluating the effect of the substance on the germination and growth of seedlings [30]. Thus, the aim of the present study was to carry out a bioassay in Petri dishes to evaluate the phytotoxic potential of estragole and 2-isopropylphenol.

## 2. Results

### 2.1. Effect of Phenolic Compounds on the Germination of Allium cepa and Lactuca sativa

In the tests carried out with *Allium cepa* on paper, the highest concentration (1 mM) of both estragole and 2-isopropylphenol showed a significantly inhibitory effect on germination rate (%GR). 2-isopropylphenol also showed a significantly inhibitory effect on germination (%GT). At the lowest concentration (0.1 mM), both compounds showed a significantly stimulating effect on these indices, whereas at 0.5 mM, no differences were observed with the control. In soil, these effects disappeared, and no significant differences were found with the control at any of the tested concentrations (Figure 1).

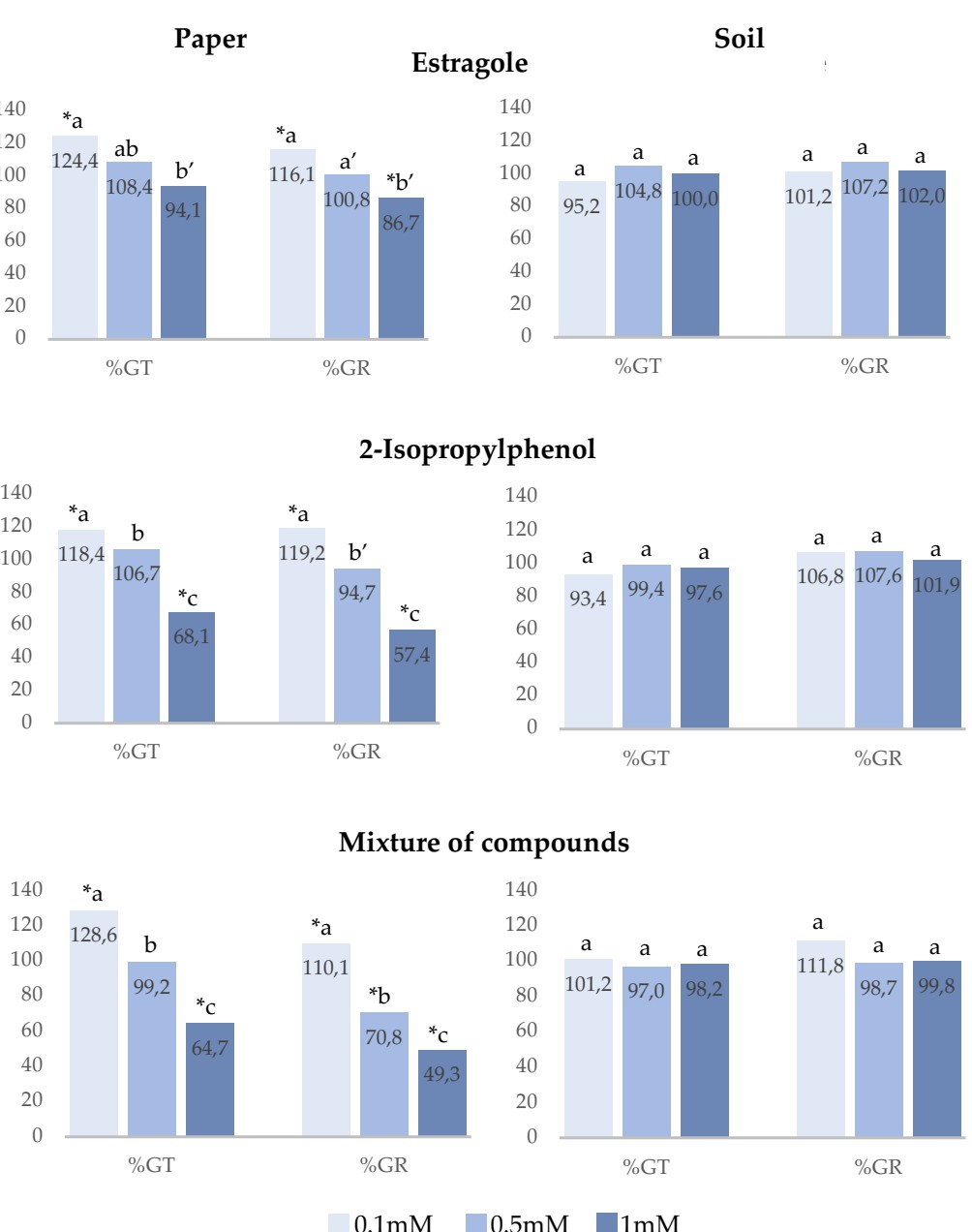

**Figure 1.** Effect of different concentrations of Estragole and 2-Isopropylphenol on *Allium cepa* germination percentage (%GT) and germination rate (%GR), expressed as the percentage relative to the control (Control value: 100%). Four replicates of each treatment were performed ($n = 4 \times 50 = 200$ seeds in total for each solution). * Significantly different from the controls. ′ Significantly different from the mixture of compounds. a, b, c: differences in small letters indicate significant differences between concentrations of the same index and for each treatment. $p < 0.05$ (Mann–Whitney U-test).

When the assay was carried out with *Lactuca sativa* on paper, 2-isopropylphenol strongly inhibited germination at 1mM, and the two compounds, separately, significantly inhibited the germination rate at all the tested concentrations, showing a significant positive correlation with concentration ($R^2 = 0.97$ for estragole and $R^2 = 0.96$ for 2-isopropylphenol). The effective concentration required to induce half maximal inhibition ($IC_{50}$) of 2-isopropylphenol on lettuce germination was 0.7 mM, and on germination rate, it was 0.1 mM for 2-isopropylphenol and 1.5 mM for estragole. As in the case of *Allium cepa* in soil, the effect disappeared, and we only observed a significant inhibition of germination rate with 2-isopropylphenol at 1 mM (Figure 2).

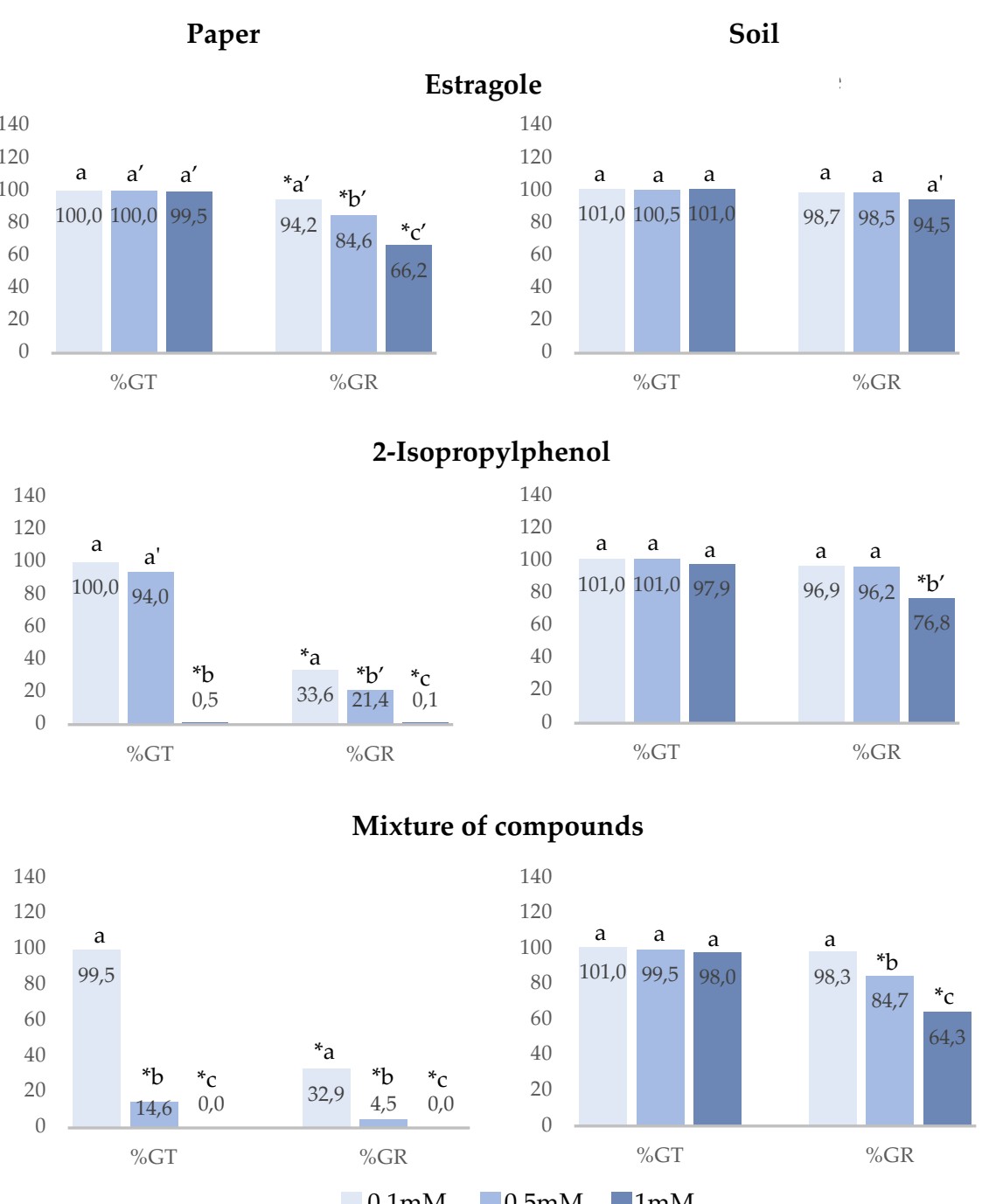

**Figure 2.** Effect of different concentrations of Estragole and 2-Isopropylphenol on *Lactuca sativa* germination percentage (%GT) and germination rate (%GR), expressed as the percentage relative to the control. Four replicates of each treatment were performed ($n = 4 \times 50 = 200$ seeds in total for each solution). * Significantly different from the controls. ′ Significantly different from the mixture of compounds. a, b, c: differences in small letters indicate significant differences between concentrations of the same index and for each treatment. $p < 0.05$ (Mann–Whitney U-test).

When tested with the mixture of the two compounds on paper, the concentration of 1mM showed total inhibition of germination in *Lactuca sativa* and inhibition of over 45% of germination in *Allium cepa*. At 0.5mM, there was also significant inhibition of germination and germination rate in *Lactuca sativa* and of germination rate in *Allium cepa*. Moreover, at 0.1 mM, there was also significant inhibition of germination rate in *Lactuca sativa*. At

0.1mM, the joint action of the two compounds showed a hormetic effect on *Allium cepa*, observing the same stimulating effect detected with the pure compounds on %GT and %GR. It is worth highlighting that the joint action of the two compounds at 0.5mM significantly inhibited %GR in *A. cepa,* and %GT and %GR in *L. sativa*, while such inhibition was not observed when the compounds were tested separately. Furthermore, in the rest of the treatments performed, this inhibition was significantly greater than that observed when these compounds acted alone. Likewise, when the test was conducted in soil, the effects observed on paper disappeared. In soil, when these compounds acted separately, they did not significantly inhibit %GR in *Lactuca sativa* at 0.5 mM, although their mixture did show a significant inhibition of %GR and, in addition, at 1mM, such inhibition of %GR was significantly greater than that observed with the pure compounds (Figures 1 and 2).

### 2.2. Effect of Phenolic Compounds on the Seedling Growth of Allium cepa and Lactuca sativa

Regarding the radicle and shoot length, when the assay was conducted with *Allium cepa,* on paper, it was observed that both estragole and 2-isopropylphenol significantly inhibited the radicle size (%Radicle length) at all the tested concentrations. These compounds also significantly inhibited the hypocotyl size (%Hypocotyl length) at 0.5 and 1 mM, although at 0.1 mM, estragole did not show any effect, and 2-isopropylphenol significantly stimulated hypocotyl growth. In soil, 2-isopropylphenol only significantly inhibited radicle and hypocotyl size at the highest concentration (1 mM). The inhibition of hypocotyl size shown by estragole on paper disappeared in soil, and the effect on the radicle was the opposite of that observed on paper, showing a significant stimulation of radicle growth that was significantly greater at lower concentration ($R^2$ = 0.98), showing an $EC_{50}$ of 0.1mm (Figure 3).

When the assay was conducted with *Lactuca sativa* on paper, estragole significantly inhibited radicle size at 1 mM and hypocotyl size at all the tested concentrations, showing a significant positive correlation with concentration ($R^2$ = 0.98). In this case, the concentration of estragole needed to inhibit the length of hypocotyl by 50% was greater than 1 mM. On paper, 2-isopropylphenol also significantly inhibited radicle and hypocotyl size at all the tested concentrations, showing a significant positive correlation ($R^2$ = 0.98 and $R^2$ = 0.97, respectively). The $IC_{50}$ of 2-isopropylphenol on lettuce radicle size was 0.4 mM, and on hypocotyl size it was 0.3 mM. When the assay was carried out in soil, the radicle size was not affected by any of the two compounds at any of the tested concentrations, and, although the inhibitory effect on the hypocotyls was lower than that observed on paper, the hypocotyl length was negatively affected by both compounds at all the tested concentrations, except for estragole at 0.5 mM (Figure 4).

When tested with the mixture of the two compounds (Figures 3 and 4), in both paper and soil, there was a significant inhibition of radicle and hypocotyl growth in *Allium cepa* at 0.5 and 1 mM, and of radicle size at 0.1 mM on paper, with the latter inhibition being greater at the highest concentration (1 mM). When the test was performed on paper, a significant positive correlation was observed with the concentration for the length of the hypocotyl ($R^2$ = 0.98) with an $IC_{50}$ of 0.6 mM. When the test was performed in soil, a significant positive correlation was also observed with the concentration for the length of the radicle ($R^2$ = 0.98) with an $IC_{50}$ of 1.1 mM. It is worth highlighting that in soil, although estragole at 1mM did not show any effect on its own on radicle and hypocotyl size, the joint action of the two compounds did, with the latter inhibition being significantly greater that that observed with pure 2-isopropylphenol. Similarly, when the assay was conducted with *Lactuca sativa*, both radicle and hypocotyl length was significantly inhibited in both soil and paper, at all the tested concentrations, except for radicle length at the lowest concentration (0.1 mM). This inhibition was greater at the highest concentration, showing a significant positive correlation with concentration for radicle length when the assay was performed on paper ($R^2$ = 0.96). In this case, the concentration of the mixture of compounds needed to inhibit the length of the radicle by 50% was 0.6mM.

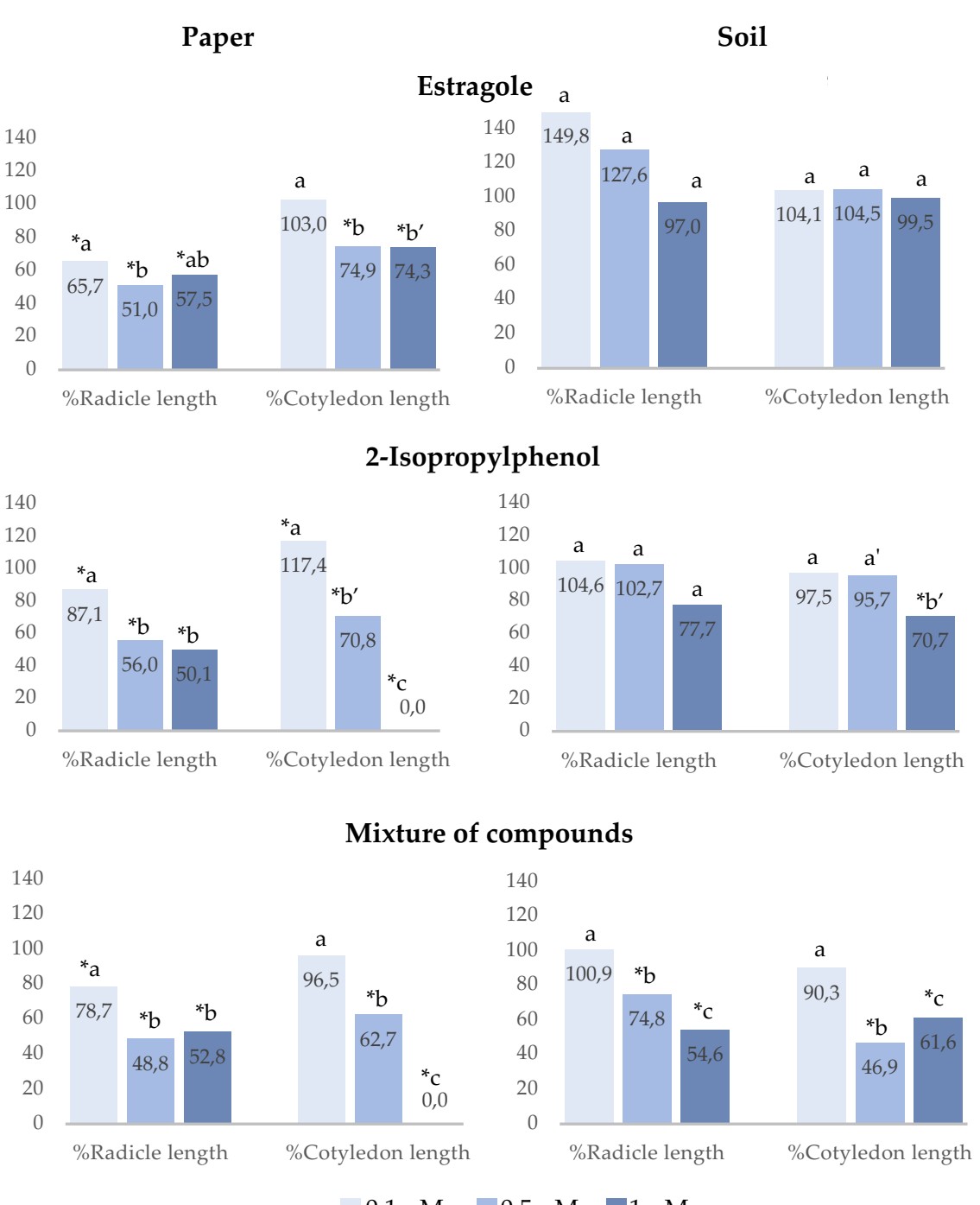

**Figure 3.** Effect of different concentrations of Estragole and 2-Isopropylphenol on the radicle and hypocotyl length of *Allium cepa*, expressed as the percentage relative to the control. Four replicates of each treatment were performed ($n = 4 \times 50 = 200$ seeds in total for each solution). * Significantly different from the controls. ' Significantly different from the mixture of compounds. a, b, c: differences in small letters indicate significant differences between concentrations of the same index and for each treatment. $p < 0.05$ (Mann–Whitney U-test).

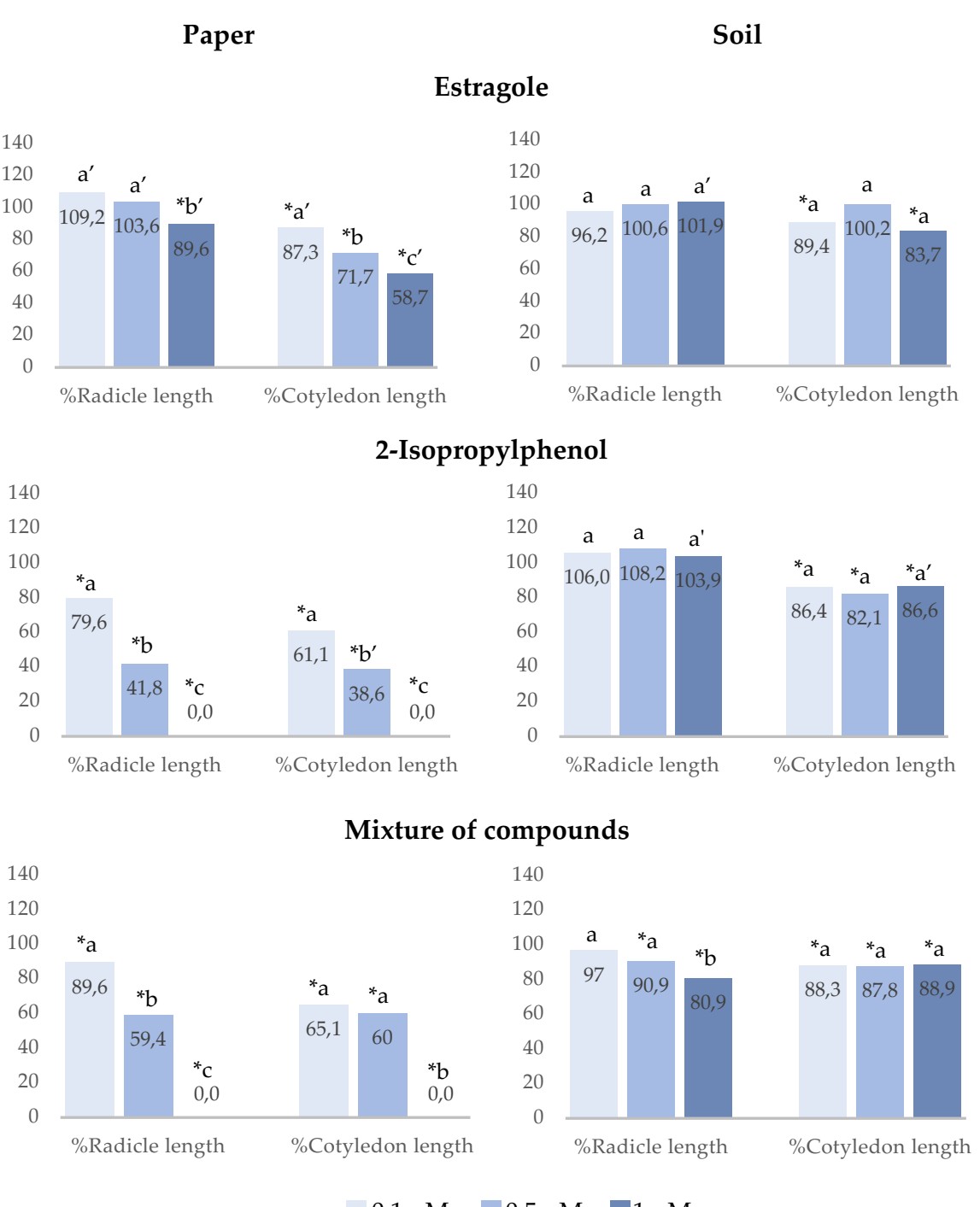

**Figure 4.** Effect of different concentrations of Estragole and 2-Isopropylphenol on the radicle and hypocotyl length of *Lactuca sativa*, expressed as the percentage relative to the control. Four replicates of each treatment were performed ($n = 4 \times 50 = 200$ seeds in total for each solution). * Significantly different from the controls. ′ Significantly different from the mixture of compounds. a, b, c: differences in small letters indicate significant differences between concentrations of the same index and for each treatment. $p < 0.05$ (Mann–Whitney U-test).

## 3. Discussion

As sessile organisms, plants cannot escape the environment in which they germinate and thus have developed a series of strategies to defend themselves against biotic and abiotic stresses. Secondary metabolites are an essential part of such strategies, as they

protect from UV rays and act as antimicrobials, repellents and anti-competitors or alle-lochemicals [31]. There are a variety of phenolic compounds of plants with allelopathic effects, many of which have shown a promising activity in weed control. Simple phenolic compounds have low molecular weight, which implies rapid degradation and prevents their accumulation in the soil and their possible influence on non-target organisms. They are water soluble, which facilitates their application without the need for surfactants [32,33]. Moreover, due to their high diversity with specific properties, they are promising tools in the discovery of new specific target places in weeds [5]. Therefore, these natural compounds can represent a large field for the discovery of new herbicides that are safe for humans, animals and the environment.

The phytotoxic activity of two phenolic compounds, viz., estragole and 2-isopropylphenol, which has never been studied before, was undertaken in this study. The results showed that these compounds presented phytotoxic activity to a lesser or greater extent depending on the substrate, the species and the tested concentration. On paper, 2-isopropylphenol at 1mM inhibited the germination of *Lactuca sativa* almost entirely. Several studies have shown that phenolic compounds can directly inhibit germination; for instance, chlorogenic acid, caffeic acid and catechol inhibit germination, as they alter the functioning of the enzyme λ-phosphorylase, which is clearly related to seed germination [5,34]. The results showed that estragole and 2-isopropylphenol also significantly inhibited germination rate and growth in the shoot and radicle of *Lactuca sativa* at the three tested concentrations on paper, observing that the higher the concentration, the greater the germination delay and the lesser the development of the seedlings. This effect on radicle growth can have equally negative consequences for any species to settle in natural conditions [15]. Previous studies show that some phenolic acids also present in *Cistus ladanifer* and at similar concentrations have the same effect on seed germination and seedling growth [7,9,10,31,35]. Pearson's determination analysis indicated that the concentration of these compounds is correlated with the tested concentrations. 2-Isopropylphenol showed an $IC_{50}$ of 0.7 mM on the germination, 0.1 mM on the rate of germination, 0.4 mM on radicle size and 0.3 mM on hypocotyl size, and an $IC_{50}$ of estragole on the germination rate was 1.5 mM and 1.1 mM on the size of hypocotyl of *Lactuca sativa*. All of this suggests that these phytochemicals can be a potential source of bioherbicides [36]. Estragole is the main compound of tarragon oil (*Artemisia dracunculus* L.) and, therefore, can be the cause of the allelopathic activity of this species.

However, when the experiment was conducted in soil, the effect was weaker, and these compounds did not affect germination directly, observing inhibition only on germination rate with 2-isopropylphenol at 1 mM. On the other hand, there was significant inhibition of hypocotyl size at the three concentrations tested. In the tests conducted in soil, the activity of compounds may have been altered due to chemical losses and transformations, which could explain why the effect in soil was weaker. The microorganisms present in the soil may also play an important role in dissipating the effect of these compounds on this substrate [37,38].

In the assay conducted with *Allium cepa*, the phytotoxic effect of these compounds was weaker than that observed on *Lactuca sativa*. In addition, it is worth highlighting that the behavior was also different. On paper, total germination, germination rate and seedling development were significantly inhibited at the highest concentration (1 mM), although at low concentration (0.1 mM), it was observed that germination and germination rate were significantly stimulated by both compounds separately; moreover, 2-isopropylphenol also stimulated hypocotyl growth at that concentration. When the assay was carried out with *Allium cepa* in soil, the effects were weaker, and significant inhibition was only observed in radicle and hypocotyl size with 2-isopropylphenol at 1 mM. It was also observed that radicle growth was significantly stimulated at low concentrations of estragole (0.5 and 0.1 mM). As was stated by Bhowmik and Inderjit (2003), the effects of one plant on another plant can be either stimulating or inhibitory depending on the concentration of the released compounds [39]. This dose-dependent activity has been observed in other phenolic compounds that are also present in *Cistus ladanifer*, such as benzoic acid and

p-hydroxybenzoic acid, with a stimulating effect at low concentrations (0.01 mM) [40]. Most allelochemicals can produce inhibitory effects at higher concentrations and stimulate growth when used at low concentrations [41]. This stimulating effect on the analyzed parameter is known as hormesis [42]. This effect can produce an average stimulation of 30–60% with respect to the control [43–45]. The application of phytotoxic compounds at low doses improves the physiological processes (including photosynthesis and respiration) and increases oxygen absorption, stomatal conductance and the water content of leaves, among other parameters [46,47]. Therefore, these compounds could also be used to enhance the growth of crop plants. Belz et al. (2009) demonstrated that, in soil, the degradation of toxic doses of allelochemicals can lead to low concentrations that may cause hormesis on plant growth [48]. The nutritional stress, together with factors such as the microorganisms, texture and physical and chemical properties of the soil, can also be one of the causes underlying the difference observed between the assay performed on paper (substrate without nutrients) and the one conducted in soil (commercial substrate with an optimal amount of nutrients).

Generally, in the phenomena of allelopathy or phytotoxic interactions among plants, several compounds are involved, and it is the combination of two or more allelochemicals that enables such activity [49]. Attacking weeds simultaneously with several allelopathic compounds with different mechanisms of action can stop resistance phenomena [28]. Thus, it is always interesting to evaluate the combination of compounds that are synthesized in the same species. In the present study, the results showed that the mixture of estragole and 2-isopropylphenol significantly inhibits germination percentage, germination rate and shoot and radicle growth in *Lactuca sativa* at the three concentrations tested on paper, observing that the higher concentration, the greater the germination delay and the lesser the seedling development. As in the case of the pure compounds, when the experiment was conducted in soil, the effect was weaker, although the mixture at 0.5 mM significantly inhibited germination rate, and at 0.5 and 1 mM it inhibited radicle size, whereas each of the compounds on their own did not show such effects. The mixture on *L. sativa* also inhibits the length of the radicle with an $IC_{50}$ of 0.6 mM. On the other hand, it was also observed that estragole stimulated the growth of the *A. cepa* radicle length in soil showing a hormetic effect with an $EC_{50}$ of 0.1 mM. When the assay was conducted with *Allium cepa* on paper, the same inhibitory and hormetic effects as those of the pure compounds were observed at high and low doses, respectively; however, when the assay was performed with this species in soil, the mixture inhibited shoot and radicle development to a greater extent with respect to the pure compounds. This is in line with numerous studies, which show that combinations of phenolic compounds present a greater phytotoxic activity compared to the same compounds on their own [7,29,31,50,51]. Mixtures of allelochemical compounds have shown a more effective inhibition of growth than each of the individual compounds separately at medium and high concentrations (0.1 and 1 mM) [40,52]. This behavior could suggest that in order for these compounds to be active, it is not necessary for them to be present at high concentrations in the medium in which they exert their action [10,53–55].

Several studies state that the extract of *Cistus ladanifer* presents a large diversity of secondary metabolites with phytotoxic activity, establishing and supporting the hypothetical relevance of allelopathy to its success in the habitats of the Mediterranean ecosystem in which it is present [8,12]. However, this proposed preliminary hypothesis of allelopathically mediated defense against other plants may not be the only rationale for all its secondary metabolites. The observation of compounds with a hormetic effect implies that some of these compounds may also play a role in growth regulation in the producing plant itself [56]. Nutrient imbalance is one of the main causes of stress in plants in natural conditions, and *Cistus ladanifer* is usually found in poor soils. Lehman and Blum (1999) reported that in the conditions of nutritional stress, the application of ferulic acid at 0.2mM increased phosphorus absorption in cucumbers (*Cucumis sativa L*) [57]. Different works identify other compounds, such as cinamic acid, in addition to ferulic, as potential stimulators of herbaceous growth [57–59]. These compounds are present in *C. ladanifer*. Therefore,

the success of this species could be due not only to a good control of competitors through the synthesis of allelopathic compounds but also to the stimulation of hormonal growth of these compounds on the individuals that constitute the rockrose population.

Studies that explored weed control using aqueous extracts of different crops found a significant increase in crop yield [60–63]. This increase could be due to a good weed control, resulting in a lower competition of the crop plants with weeds, although it could also be due to the stimulation of the hormonal growth in the crop plants. Thus, although the secondary metabolites of plants continue to be a potential source for the development of natural herbicides, it is also necessary to pay attention to the hormetic effects that they may present at low concentrations. Most studies on phytotoxicity are conducted in laboratory conditions to remove other factors derived from the properties of the soil. Such approach allows recognizing only the direct effects of allelochemical action. There is still great need for transferring the laboratory data to the field conditions. In fact, the concentrations at which the allelochemicals are found in field conditions are usually lower than those at which these compounds showed phytotoxic activity. In the framework of a wider selection aimed at identifying more environmentally friendly tools and management strategies for the creation of a more sustainable agriculture, the hormetic effect of allelopathic compounds may have a greater practical potential compared to their inhibitory potential. It would be ideal to find allelochemicals that, when applied in the field at the beginning of the cultivation process in sufficient amounts to control the growth of weeds, would then degrade to doses that improve the crop yield. On the other hand, it should be noted that the application of high concentrations of any compound in the soil can be harmful to the environment and human health. The exposure to estragole resulting from consumption of herbal medicinal products (short-time use in adults at the recommended posology) does not pose a significant cancer risk, but at high concentrations, it can be carcinogenic and genotoxic [64].

From the present study, we can conclude that the effect of the tested allelochemicals can vary depending on the substrate, the dose and the tested species, therefore corroborating with other studies [65,66]. For this reason, it can be pointed out that in the study of allelopathy or the search for new herbicides, in order to draw conclusions about what really occurs in the ecosystem in which the species is found or to know what will occur in the cultivated soil where the allelochemical will be applied, it is necessary: (a) to carry out studies with different species, in order to clarify the specificity of the compound; (b) to use a range of concentrations that show possible phytotoxic and hormetic effects; (c) to reproduce in the experiment the physical–chemical and biological properties of the natural environment; (d) to approach the study while taking into account the presence of other compounds that can enhance the activity of these compounds. In addition, all of this, and especially the last point, can help in finding alternatives to the phenomenon of weed resistance.

## 4. Materials and Methods

### 4.1. Plant and Substrate Sources

We analyzed variables such as the effect of the compounds, separately and in combination, on two receptor species: *Allium cepa* and *Lactuca sativa*, representatives of mono- and di-cotyledoneae plants. These species are ideal for adequately showing the allelochemical effects on the germination processes [67,68]. Moreover, *Lactuca sativa* is recommended by the United States Environmental Protection Agency (US EPA) for phytotoxicity tests, being among the most sensitive species [69]. *Allium cepa* has proved to be the most useful and has repeatedly been suggested as a standard test material in the list of species historically used in plant testing by the International Organization for Standardization [70] and by the Organisation of Economic Cooperation and Development [71].

To carry out the bioassay, seeds of *Lactuca sativa* variety romaine verte maraîchère (Vilmorin Jardin—CS70110—38291 St Quentin Fallavier Cedex—France) and *Allium cepa* variety bianca di maggio (Vilmorin Jardin—CS70110—38291 St Quentin Fallavier Cedex—

France) were used as representatives of dicotyledons and monocotyledons, respectively. The total germination of these seeds was over 98%.

The activity of the allelochemicals was carried out on Whatman No. 118 paper and a commercial substrate of universal type prepared based on 95% peat, 5% green compost and 1.3 Kg/m$^3$ fertilizer: 12N + 12P + 17K (Geolia, Aki Bricolage España S.L. B-839857—Madrid, Spain). The commercial universal soil presented the following characteristics: organic matter per dry matter (60%), electrical conductivity (40 mS/m), apparent dry density (320 g/L), grain size (0–20 mm) and pH 5.5–6.5.

The activity of the allelochemicals can vary according to the planting substrate. For this reason, the experiment was carried out on Whatman No. 118 paper and on a commercial substrate. The commercial universal soil presented the following characteristics: organic matter per dry matter (60%), electrical conductivity (40 mS/m), apparent dry density (320 g/L), grain size (0–20 mm) and pH 6.5.

None of the materials (seeds and substrate) was sterilized before the experiment.

*4.2. Phytotoxic Activity Test*

Chemically pure reagents (estragole and 2-isopropylphenol of >98% purity) were obtained from Aldrich-Chemical. Different solutions were prepared with Milli-Q water from each component separately. The highest concentration was 1 mM, which is the maximum recommended concentration in allelopathic bioassays [35]. The minimum concentration was 0.1 mM, which was the lowest concentration for observing the dissipation of the effect of these compounds. In addition, solutions of 0.5 mM were also prepared. For the mixture of the two phenols, three solutions were prepared with equimolar concentrations of each of the compounds at 1 mM, 0.5 mM and 0.1 mM. To eliminate the effects of pH, we measured these parameters for each solution. The pH varied between 6.1 and 6.3 from one solution to another. There were no significant differences in pH among solutions.

In Petri dishes (four replicates for each experience), 50 seeds (200 in total) of *Lactuca sativa* or *Allium cepa* were placed on paper or 25 g of commercial substrate and watered with the different solutions prepared (5 mL for paper plates and 16 mL for soil). Control plates were watered with Milli-Q water. The plates were kept in a culture chamber at 22 °C for a photoperiod of 15 h of light and 9 h of darkness (5 days for *L. sativa* plates and 6 days for *A. cepa* plates).

*4.3. Measured Indices to Quantify the Phytotoxic Effect*

Germinated seeds were counted daily, and total germination (%TG) [7] and germination rate (%GR) [69,70] were calculated. On the last day of the experiment, 10 seedlings were chosen at random, and the length of the radicle and shoot was measured [72]. All results were expressed as a percentage relative to the control.

*4.4. Statistical Analysis*

The significance level of the comparisons among treatments was estimated using the Mann–Whitney U-test. The differences were considered significant when $p < 0.05$. The interrelationships between germination and seed growth with the concentration of phenolic compounds were determined by Pearson's determination coefficient. The effective concentrations required to induce half maximal inhibition (IC$_{50}$) or half maximal stimulation (EC$_{50}$) of growth were calculated according to the linear relationship between concentration and per cent inhibition or stimulation of plant growth. All statistical analyses were conducted using the statistical software SPSS 15.0.1.

**Author Contributions:** Conceptualization, T.S.; methodology, E.R.; validation, T.S., A.d.R.S. and D.O.; formal analysis, E.R.; investigation, E.R. and T.S.; resources, T.S. and A.d.R.S.; data curation, D.O.; writing—original draft preparation, T.S.; writing—review and editing, E.R.; visualization, A.d.R.S. and D.O.; supervision, D.O.; project administration, A.d.R.S.; funding acquisition, T.S. and A.d.R.S. All authors have read and agreed to the published version of the manuscript.

**Funding:** This research was funded by the Regional Government of Extremadura and the European Regional Development Fund, grants number GR18078 and IB18105.

**Data Availability Statement:** The data presented in this study are available on request from the corresponding author.

**Conflicts of Interest:** The authors declare no conflict of interest.

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
