# Peer review of "Evaluation of the Activity of Estragole and 2-Isopropylphenol, Phenolic Compounds Present in Cistus ladanifer"

_agronomy, doi:10.3390/agronomy12051139_

Round 1

Reviewer 1 Report

I hope this is good piece of work but needs major revision as it is my first impression of this paper, so that it should be made more clear. Please notice my comments stated below:

Plagiarism part should be checked by the officials of the journal.

In case of Abstract page 1, line 11, what does a point to phenolic mean, it does not suit here.

Line 14, it is grammatically wrong, correct it.

Line 19, Germination speed should be relaced at every place as germination rate..

Line 21: When assay are done on soil are disspiated what does it mean and replace 0.6 mm by 0.6 mM.

Line 25 What does an hose effect mean and also replace 0.1 mm by 0.1 mM

Over all the abstract needs meaningfull and crisp writing.

Introduction

Page2

 Line 59,  Change the text of the line

Line 66/67 need change of text for clear understanding.

Line 70, This compounds should be replaced by this compound.

Line 73 -80, set it right in a crisp manner.

In case of Results

 Page 3,

Use instead of term rate instead of speed  

In case of table 1, keep text of table legend only up to germination rate and other part of the legend should be placed at the bottom of the table. Please follow same trend for other three tables.

Page 4 line 158 place 0.1 mm as 0.1mM

In case of discussion Page 5 line 201, please  use they germinate

Page 6 line 218 almost entirely should be replaced by a technical term

Lines 249-254, need explaination.

 Last para of Discussion needs crisp and meaningfull writing.

Over all Discussion is a too big need about 50% reductions that too with mechanisms.

Methodology

 Page 9 Line 367, experiment instead of experience

Line 369 what is the value of electrical conductivity.

Line 375 Replace minor with minimum,

Line 377-379, since  the concentations are different like 1 mM, 0.5 mM and 0.1 mM. How can these be equimolar?

Line 384, replace control dishes with control plates.

Author Response

We thank the reviewer for his/her vote of confidence.  The specific comments were repaired. See attached manuscript.

Reviewer 2 Report

The manuscript entitled "Evaluation of the activity of Estragole and 2-Isopropylphenol, 2 phenolic compounds present in allelopathic species” by Requesón et al., describes the activity of two phenolic compounds, Estragole and 2-isopropylphenol on paper and soil against Allium cepa and Lactuca sativa. The authors highlighted a significantly different effect substrate, dose and species dependent of the tested allelochemicals. The topic is of potential interest for the readership of the journal and, in general, the data are well presented and of good quality. Just some questions need to be answered and few details should be corrected to improve the manuscript quality and clarity.

Materials and Methods:

Page 9, line 370: Was the soil sterilized before the use? Please improve this part.

Page 9, line 386: Did the authors the same phytotoxic activity test on soil too, as reported in the results? Please, explain the methods used with soil too.

Page 9, line 388: What it means “[Cristina]”?

Results:

Tables 1,2, 3 and 4: Please, transform tables into graphs to better explain the results achieved.

Discussion:

Page 7, line275: Please, add some citations.

To improve the discussion, please answer to the following questions:

If the soil was not sterilized, can the miocroorganisms present influence the allelochemicals effects?

Can the authors better explain the possible interaction between Estragole and 2-Isopropylphenol to better understand the higher activity pf the compounds mixture observed?

Author Response

(The authors gave the same response as above.)

Reviewer 3 Report

Comments and Suggestions for Authors

The present manuscript describes an interesting approach to evaluating the impact of estragole and 2-Isopropylphenol, 2 phenolic compounds present in Cistus ladanifer on Allium cepa and Lactuca sativa plants. However, more detailed explanations of experimental conditions and careful reconsideration of certain arguments in the discussion are needed before the manuscript is ready for publication. Most parts of the abstract, introduction, material and methods, results, and conclusion must be rewritten. In this form, the text is very difficult for reading. English language editing and style polishing for this manuscript is necessary. The authors must be making a MAJOR REVISION and resubmitting the manuscript.

Title

Line 1: I do not agree with this " allelopathic species". The proposal for a new title is: Evaluation of the activity of estragole and 2-Isopropylphenol, phenolic compounds present in Cistus ladanifer

Abstract

Line 10-15: Please replace the part from lines 10 to 15 with the following text: A large number of studies of Cistus ladanifer highlight this Mediterranean shrub as a source of the phenolic compounds responsible for the allelopathic potential of this species. There are few phenolic compounds present in C. ladanifer that have not yet been studied. The objective of this work is to evaluate the activity of estragole and 2-isopropylphenol on filter paper and soil on monocotyledons (Allium cepa) and dicotyledons (Lactuca sativa).

Line 16-24: This section is with too many wordy and unclear sentences. Please, rewrite this part.

Line 25-28: I do not understand what you concluded. What do you mean as you say "normal concentrations"? Please clarify.

You need to write only the most important data which you observed in your study. In according you need to write a take-home message, about why your results are important.

Finally, the complete text of the abstract must be rewritten.

Keywords: Please delete „interaction of phytotoxic compounds“.

Introduction

Line 87: Please replace with: Also, this study.....

Line 90-97: This part removes in the M&M section.

Line 98: Please check in the author's guidelines, which section is first, M&M or Results.

M&M

Line 362: Please replace the subheading „Selection of seeds and substrate“ with „Plant and substrate sources“

Line 363-370: „To carry out the bioassay, seeds of Lactuca sativa and Allium cepa were used as representatives of dicotyledons and monocotyledons, respectively. These seeds were obtained commercially and showed total germination of over 98% The activity of the allelochemicals can vary according to the planting substrate, for this reason the experience was carried out on Whatman No. 118 paper and on a commercial substrate. The commercial universal soil presented the following characteristics: organic matter per dry matter (60%), electrical conductivity (mS/m), apparent dry density (320 g/L), grain size (0–20 mm) and pH 6.5.“

Please replace it with this text:

„To carry out the bioassay, seeds of Lactuca sativa (please add variety and producer) and Allium cepa (please add variety and producer) were used as representatives of dicotyledons and monocotyledons, respectively. The total germination of these seeds was over 98%. The activity of the allelochemicals was on Whatman No. 118 paper and a commercial substrate (which one?). The commercial universal soil presented the following characteristics: organic matter per dry matter (60%), electrical conductivity (mS/m), apparent dry density (320 g/L), grain size (0–20 mm), and pH 6.5.“

Line 371-386: Phytotoxic activity test: In this subheading also you should correct your English. The sentences are too wordy and grammatically incorrect.

Line 388: Cristina?

Line 389: Why S? Germination rate (GR) or germination speed (GS). Please, add formulas for both parameters.

Line 389-390: Please delete this sentence: These compounds can directly inhibit germination, but can also affect seedling development.

Line 390-392: Therefore, at the end of the experiment, 10 seedlings were chosen at random and the length of the root and stem was measured [72]. All results have been expressed as a percentage relative to the control.

Please replace it with: Last day of the experiment, 10 seedlings were chosen at random and the length of the radicle and shoot was measured [72]. All results have been expressed as a percentage relative to the control.

Radicle and shoot are more appropriate terms for seedling than root and stem. Please change them throughout the text.

Results

Line 100-107: The sentence is too long and unclear, like most others in the results section. It is undesirable to start a sentence with "when". Why did you choose to present the results in Tables in this way? In addition to the fact that it is not common to present them in this way, it is very unclear. The tables should show the real values ​​obtained by measuring the parameters. After that, the result section should comment on the percentage of inhibition or stimulation. You did not put the recalculated percentages of inhibition or stimulation in the tables, the reader should do it himself. I find this to be a very inappropriate way of presenting results.

In the M&M section, you wrote that the germination of the commercial seed was over 98%. This germination was in control? How is it possible that in Table 1 for the concentration of 0.1mm you got the values 124.4, 118.4, 128.3, respectively? There was the same number of seeds in all the treatments, how is it possible that there is so much stimulation?

FINALLY, THE COMPLETE RESULTS MUST BE RECALCULATED, ALSO THE TEXT OF THIS SECTION AND DISCUSSION MUST BE REWRITTEN.

Author Response

(The authors gave the same response as above.)

Round 2

Author Response

Dear reviewer,

The authors have carried out all the suggested changes.

Thank you for your time on the new review.
